# Virus-like Particles: Measures and Biological Functions

**DOI:** 10.3390/v14020383

**Published:** 2022-02-14

**Authors:** Tara Bhat, Amy Cao, John Yin

**Affiliations:** Department of Chemical and Biological Engineering, Wisconsin Institute for Discovery, University of Wisconsin-Madison, 330 N. Orchard Street, Madison, WI 53715, USA; tarambhat@gmail.com (T.B.); ahcao@wisc.edu (A.C.)

**Keywords:** virus-like particles, defective interfering particles, semi-infectious particles, cell killing particles, defective viral genomes, plaque forming unit, multiplicity of infection, co-infection, transmission electron microscopy, clonogenic assay, coulter counting, resistive pulse sensing, flow virometry

## Abstract

Virus-like particles resemble infectious virus particles in size, shape, and molecular composition; however, they fail to productively infect host cells. Historically, the presence of virus-like particles has been inferred from total particle counts by microscopy, and infectious particle counts or plaque-forming-units (PFUs) by plaque assay; the resulting ratio of particles-to-PFUs is often greater than one, easily 10 or 100, indicating that most particles are non-infectious. Despite their inability to hijack cells for their reproduction, virus-like particles and the defective genomes they carry can exhibit a broad range of behaviors: interference with normal virus growth during co-infections, cell killing, and activation or inhibition of innate immune signaling. In addition, some virus-like particles become productive as their multiplicities of infection increase, a sign of cooperation between particles. Here, we review established and emerging methods to count virus-like particles and characterize their biological functions. We take a critical look at evidence for defective interfering virus genomes in natural and clinical isolates, and we review their potential as antiviral therapeutics. In short, we highlight an urgent need to better understand how virus-like genomes and particles interact with intact functional viruses during co-infection of their hosts, and their impacts on the transmission, severity, and persistence of virus-associated diseases.

## 1. Introduction

When multiple observations or measures can be made on the same or related phenomena, their combination or comparison may lead to new understanding or insights. In astronomy, for example, multiple measures of planetary positions across seasons provided compelling evidence for a universe that was heliocentric rather than geocentric [1]. In biology, the diversity of finches observed within and between different islands provided evidence for Darwin’s theory on the origin of species [2]. In virology, the development of foci-forming [3] and plaque-forming assays [4] enabled the quantitative measures of infectious virus particle levels as foci-forming or plaque-forming units (PFUs); when such measures were combined with estimates of total virus-like particles, initially employing electron microscopy [5], the tally of particle-to-PFU ratios that differed from unity showed that most virus-like particles were not infectious. For example, particle-to-PFU ratios have ranged from 1-or-2 for bacteriophages and vaccinia virus [6] to 10 for herpes virus [7], 20-to-50 for influenza and 30-to-1000 for poliovirus [8], to 40,000 for varicella zoster virus [9]. The ratio for SARS-CoV-2 ranges from 10^4^-to-10^6^ based on RNA as a proxy for total particles (genomic RNA-to-PFU) [10,11]. Although virologists are aware that different viruses can exhibit very different particle-to-PFU ratios, it is not so widely known that non-infectious virus-like particles can exhibit a diversity of biological activities. We focus here on non-infectious virus-like particles that arise as byproducts of virus growth in susceptible cells, which can be found in Nature or from laboratory cultures; we do not review here the extensive literature on engineered virus-like particles for applications as subunit vaccines [12] or drug delivery vehicles [13]. By reviewing methods for the detection and quantitative characterization of virus-like particles, we aim to spotlight their diverse functions and activities.

## 2. Counting Particles

### 2.1. Virus and Virus-like Particles

The counting of virus and virus-like particles of a laboratory, environmental, or clinical sample exploits physical, chemical, or biological aspects of the virus. These include the size, shape and electrical conductivity of virus particles, their propensity to be specifically labeled with one or more fluorophores, and the ability of the virus particle to infect a host cell, cause cell death, or produce virus progeny. Chemical and physical methods of particle quantification are summarized in Table 1 [5,14,15,16,17,18,19,20,21,22,23].

#### 2.1.1. Transmission Electron Microscopy

Transmission electron microscopy (TEM) has been used to identify, classify, and quantify different viruses based on morphology [16]. Historically, the first virus visualized by TEM was orthopoxvirus [5]. Other discoveries include adenovirus, enterovirus, paramyxovirus, and reovirus. TEM imaging has been crucial in diagnosing viruses such as smallpox and chickenpox [24], as well as identifying emergent diseases and outbreaks such as West Nile [25] and severe acute respiratory syndrome (SARS) [26]. More recently, the novel human coronavirus associated with COVID-19 was confirmed by its morphological features under TEM visualization [27].

TEM focuses an accelerated electron beam upon a thin specimen that transmits an image onto a screen. It has a higher resolution than light microscopy, owing to the shorter wavelength of electrons (0.1 angstrom). Thus, TEM can capture atomic scale resolution of viruses, which are typically 20-to-200 nm in size. Before the establishment of electron microscopy for determining particle-to-PFU ratios, the first ratio between the count of infectious particles and the count of physical virus particles was measured for vaccinia virus, with the quantity of “elementary bodies” being calculated using the dry weight of a sample [28]. Elementary bodies were first observed as micron-sized particles from fowl pox lesion scrapings, later realized as components of the virus particles. The elementary bodies can be isolated by centrifugation and sedimentation, and the dry weight of the elementary bodies is divided by the calculated mass of one elementary body. This calculation allowed for error in the estimation, reflecting sample heterogeneity. The plaque assay was used to determine the infectious titer of the sample; for vaccinia, this resulted in an average ratio of 4.2 particles per infectious unit [28]. Years later, the particle-to-PFU ratio for vaccinia was found to be 1.5 particles per infectious unit using electron microscopy [6].

For virus quantification, samples are often mixed with latex beads of known size and concentration [29,30]. Manual counting is conducted to quantify the number of virus particles and beads in several view fields and using the relative counts of virus particles and stock beads, the titer of virus in particles/mL can be determined. Defective particles can, in some cases, be detected by TEM, owing to differences in their size and morphology from wild-type particles (Figure 1); defective interfering particles of vesicular stomatitis virus (VSV) are often visibly truncated, with a rounded shape of diameter 76 ± 8 nm, which is shorter than the prototype bullet-shape 70 ± 8 nm by 204 ± 14 nm of wild-type VSV particles [31].

While TEM can produce high-resolution images, its throughput is limited by its field of view. Higher magnification enables greater confidence in particle identification, but the sampling field is then smaller. Interpretation of TEM images depends on the operator, and during virus quantification the accidental counting of debris can cause overestimation. Owing to its dependence on human counting, TEM quantification has a relatively low throughput. Particle counting requires a high enough number of particles to reflect the average number in the sample, but not so high as to overwhelm the manual counter. In practice, such counting has a lower detection limit of 10^7^ particles per mL; samples may need to be concentrated prior to analysis. Specimen preparation can be complex and tedious, typically requiring several hours, and artifacts in the images may result from the preparation steps. The virus is purified from cell debris, and a suitable supporting film needs to be prepared to hold the sample, often using a coated copper grid [29]. Three methods have been compared, employing sucrose-density purification followed by negative staining, thin section electron microscopy of pelleted resin-embedded supernatants, and direct counting after negative staining [30]; direct counting, where a known titer of latex beads was added to samples, was found to be the most accurate and reproducible.

#### 2.1.2. Epifluorescence Microscopy

In epifluorescence microscopy (EFM) virus particles are stained with fluorescent dyes that can bind nucleic acids and proteins of the virus. Fluorescence microscopy uses dyes that will produce a distinguishable signal, with the requirement being a signal-to-noise ratio higher than five [19]. DAPI, Yo-Pro-1, SYBR Green, and SYBR Gold are dyes commonly used to stain nucleic acids [18,32]. Proteins are specifically labeled by monoclonal antibodies (MAb); typically, the MAb or primary antibody is further bound by a secondary antibody that is covalently linked with a fluorescent dye. When exposed to different excitation wavelengths of light, different dyes will fluoresce. Conventionally, fluorescence microscopes are limited by the diffraction of light, with a resolution of 200–350 nm [33].

Compared to TEM, EFM equipment is less expensive, and sample preparation is relatively simple, making EFM a favored method for analyzing samples in the field. EFM has been most commonly used to count viruses in marine samples. Collecting and fixing samples takes 15–20 min, slide preparation takes one hour, and enumeration takes 30 min [34]. The EFM method using nucleic acids stained by Yo-Pro or DAPI were more precise than the TEM method, with TEM underestimating the number of viruses in samples [35,36]; counts of SYBR Gold-stained viral particles using direct epifluorescence microscopy were 1.34 times higher than counts based on TEM [17].

#### 2.1.3. Resistive Pulse Sensing

Resistive pulse sensing (RPS) is a particle detection method based on the Coulter principle, first developed and released by Wallace Coulter more than sixty years ago [37]. The principle takes advantage of the low electrical conductivity of the cells or particles relative to the conductive aqueous salt solution in which they are suspended; it refers to a change in impedance comparable to a particle’s volume when it passes through a pore, momentarily reducing the flow of electric current through the solution, called a blockade event. The pore of the device is large enough for the particle to pass through, but small enough that the current flow through the cross-sectional area of the pore is detectably perturbed by the passage of the particle. The device is calibrated by a solution containing nanoparticles of known size and concentration; an appropriate dilution is needed to minimize the possibility that two particles simultaneously occupy the pore. Rod-shaped virus particles such as Tobacco mosaic virus may rotate in order to pass through a solid-state nanopore, which produces noise and unclear signals in the current readings [20,38]. For virus populations with varied sizes and aggregates, particles may become stuck in the pore. This can be avoided by adjusting the nanopore size or by using tunable elastic nanopores.

Tunable resistive pulse sensing (TRPS) is a variant of RPS techniques that uses an elastic membrane containing a pore that can be stretched or relaxed, differing from the standard fixed solid-state pores. The resulting size of the pore aperture can be fine-tuned to better match different sizes of particles, which is especially important for polydisperse populations of viruses that have a wide range of sizes. The ability to determine the size of blockade events can also allow for the determination of coincidence and aggregate events, for example, when the pulse magnitude is an integer multiple of the typical individual pulses [39]. The pulse shape depends on the trajectory and movement speed of the particles. The stretchability of the pore also allows for recovery from blockages [40,41], to prevent clogging. This technology was mainly developed by Izon Science [20], and their qViro platform is able to provide estimates for the concentration of particles, as well as the size distribution and surface charges of particles [42]. Currently, the smallest commercial nanopores have a diameter of 100 nm, with a minimal detection limit of 70 nm [20,42]. In practice, TRPS has a quantification range from 10^7^ to 10^10^ particles/mL [21].

#### 2.1.4. Flow Cytometry and Virometry

Flow cytometry combined the Coulter principle with fluorescence detection as a high-throughput method to measure the nucleic acid content and size of cells [43]. Cells are directed one-by-one into a flow stream by a laminar sheath-flow system, and they are excited by a laser beam. The resulting deflected light is characterized as forward scatter (FSC), measuring size of the cell, and side scatter (SSC), reflecting its granularity, a measure of surface irregularity or coarseness. It is noteworthy that flow cytometric measures of infected-cell granularity have been found to correlate with resulting virus titers for herpes simplex virus [44]. A standard flow cytometer is not able to effectively detect particles below 500 nm; modifications that include the use of high-wattage excitation lasers and reduced flow chamber diameter [45], wider-angle sampling of scattered light [46], as well as fluorescent labeling, have enabled virus particle quantification [47,48]. Based on its origins from flow cytometry, this technology has been called flow virometry. A schematic for flow virometry is shown in Figure 2.

A powerful feature of flow virometry is the ability to characterize heterogeneity at the level of individual virions. By taking advantage of multiple fluorescent labels, a flow virometer can characterize multiple characteristics of individual virus particles. For example, magnetic nanoparticles (MNPs) coupled with specific monoclonal antibodies have been used to capture and separate virions, and the separated complexes have been analyzed by flow virometry. This technique has been used to analyze the maturation of Dengue virus particles produced from different sources [49], and the antigen and envelope protein composition of HIV-1 particles [50,51]. Current limitations of MNP capture are that only virions complexed with MNPs can be analyzed and steric interference, which limits the antigens per virion that are accessible for labeling by antibodies. Many modern flow virometers are able to sort heterogeneous samples using fluorescence-activated cell sorting (FACS), which allows for further characterization of the sorted particles. A flow virometry assay was used to characterize the RNA and glycoprotein content of Junin virus particles that were sorted and analyzed for distinct infectivity profiles based on size [52].

Because flow virometry is flow-based and non-visualization based, flow virometry is a fast technique that relies less on the operator than EFM, with throughput up to 2000–6000 particles per second and producing results for a sample in less than 1 h. Operating a viral flow cytometer is less technically demanding than other methods such as TEM, and many steps of the process are automated [22,48]. Disadvantages of flow virometry include the potential for overestimation of particle levels due to background noise. Events of coincidence will underestimate particle counts, which is why determining a correct dilution is important for flow-based analysis. The probability of coincidence events can be described by the Poisson distribution:$$P\left(n\right)=\frac{{\left(rt\right)}^{n}e^{-\left(rt\right)}}{n!}$$
where *r* is the flow rate (particles per second) through the detection volume and *t* is the time spent within the detection volume. Typically, a high concentration of about 6 × 10^8^ particles/mL gives 10 percent probability of two particles occupying the detection volume at the same time, with higher concentrations resulting in greater probability of coincidence. The lowest limit of detection of flow virometry is around 100 nm [48], limiting the detection of smaller virus particles.

### 2.2. Infectious Virus Particles

Infectious virus particles are quantified by plaque assay or end-point dilution, which both involve visualization of macroscopic regions of cytopathology. Virus particles that infect cells and kill them without making virus progeny can be quantified by their ability to prevent formation of cell colonies, called the clonogenic assay. These methods are summarized in Table 2 [8,53,54,55,56].

#### 2.2.1. Plaque Assay

The plaque assay is the most widely used method for the quantification of infectious particles or virus titers. The assay is carried out by preparing serial dilutions of a virus stock of unknown titer and applying them to susceptible cell monolayers. After adsorption and infection initiation, cells are overlaid with agar to localize the spread of subsequent rounds of infection to the vicinity of initial infected cells; macroscopic regions of cell death called ‘plaques’ can be made visible by crystal violet, which stains intact cells and leaves dead-cell or infected areas unstained. Since each plaque arises from the amplification of an initial single infectious particle, plaque counts, combined with known volumes of known dilutions can be used to calculate the concentration of infectious particles in the stock sample. The infectious virus titer is reported in plaque forming foci (PFF) or plaque forming units (PFU) per ml of solution.

#### 2.2.2. End-Point Dilution

An alternative to the plaque assay, the end-point dilution assay, is also used to quantify infectious virus titer. End-point dilution measures viral titer based on the dilution at which half of the total cell cultures become infected, expressed as a 50 percent tissue culture infectious dose (TCID_50_) per mL. A virus stock of unknown titer is used to produce 10-fold dilutions, where one milliliter of each dilution is applied to multiple (say ten) cell cultures. After incubation, plates are inspected for cell death or cytopathic effects (CPE). The dilution at which 50% of the cultures exhibit CPE is considered the end point; for example, if 50 percent of the cultures at 10^4^-fold dilution exhibit CPE, then the stock has a titer of approximately 10^4^ TCID_50_ per ml. In practice, the TCID_50_ is determined from this dilution by calculation using the Spearman–Karber or Reed–Muench method, though other methods exist [57,58]. Values of TCID_50_/mL and PFU/mL are not equivalent, but are comparable [57]. For consistency and owing to the broader use of the plaque assay, we focus here on PFU or PFF measures of virus titer.

### 2.3. Cell-Killing Particles (Clonogenic Assay)

The clonogenic assay, also known as the colony forming assay, was first developed by Marcus and Puck to determine the effects of radiation on cells [59]. The procedure was later adapted to measure cell-killing particles (CKPs) of Influenza A [55]. In a clonogenic assay, cells are grown in monolayers. Virus particles are attached to the monolayers at various known multiplicities of plaque-forming particles (PFP), and the infected cells are monodispersed and seeded into culture plates to allow for colony formation. After colonies are produced, they are fixed and stained for counting [60]. A survival curve is produced from the fraction of surviving cell colonies remaining from each multiplicity of PFP. Using the Poisson distribution with the survival curve function, the titer of CKPs can be calculated assuming that the virus attachment is nearly 100% and that each CKP will kill the cell it infects, resulting in no visible colony [61]. Marcus and Sekellick first used a cell-killing assay to measure ratios between CKPs and PFPs in three different peaks of DIP and PFP activity [61]; more recently the clonogenic assay was used to show for a variant of influenza A virus that CKPs were seven-fold more prevalent than PFPs [55].

## 3. Virus-like Particles: Emergence, Function, and Prevalence

Although they are often dismissed as bothersome non-infectious byproducts of standard or infectious virus cultures, virus-like particles can exhibit diverse biological functions; these include interference with normal infection, induction of apoptosis or host-cell killing, and activation of innate immune signaling. While next generation and single-molecule sequencing have revealed the heterogeneity of natural and clinical virus isolates, both in genome sequences and lengths, studies have focused on subpopulations that carry full-length genomes and are infectious. Understanding how genetic variation contributes different functions in different cellular and infection environments remains an aspect of natural virus populations that is largely unexplored. Understanding how to quantify the functional diversity of virus-like particles may provide insights into their ecological and evolutionary roles in the natural persistence of viruses and suggest more robust strategies for their management. Below we discuss the different virus-like particles and how to characterize their biological functions.

### 3.1. Defective Interfering Particles

Virus particles that are unable to replicate independently are called defective or non-infectious particles. The most widely studied class of defective particles can interfere with normal or standard virus infections, as shown in Figure 3. Defective interfering particles, often abbreviated as DI particles or DIPs, were first characterized by Preben von Magnus, who discovered that mice inoculated with the second and third culture passages of influenza virus exhibited few signs of infection; incomplete influenza virus particles were interfering with wild-type replication [62]. Their name, defective interfering particles, coined in 1970, reflects both the non-infectious but function of these particles [63].

DIPs and their associated defective genomes were useful probes of replication mechanisms, before and during the rise of recombinant DNA technologies in the 1980s and 1990s. Cloned cDNA samples from vesicular stomatitis virus (VSV) and other RNA viruses were used to recover DIPs [64], and polymerase chain reaction assays were developed in 1992 to measure copy-back and other defective viral genomes [65].

#### 3.1.1. DIP Emergence

DIPs arise as byproducts of virus replication and infection. When a cell is infected by a virus, the viral replication complex uses the virus genome or anti-genome as a template to instruct the synthesis of full-length genomic templates for eventual packaging into virus progeny particles. Stretches of the template that are flanked by similar or identical short (less than 20 nucleotide) repeat sequences can enable hopping of the elongating replication complex between the repeats, causing within-strand or between-strand recombination, leaving a deletion in the resulting genomic template [66,67,68]. If the deletion or other mutation causes the loss of an essential virus function, then the replication product is functionally defective; the product is a DVG or defective viral genome. DVGs have long been associated with DIPs. More recently, DVGs have been discovered in natural and clinical virus isolates, enabled in part by advances in deep and single molecule sequencing. Interest in the biological functions of DVGs is growing, especially regarding their possible roles in disease development and severity; the details of which are reviewed elsewhere [69,70]. Two facets of DVGs that contribute to their persistence in nature have to date garnered less attention: their emergence and potential for evolution.

The structure and evolution of DVG populations will depend on their rates of generation. Here, it is useful to distinguish between rates and frequencies of mutation (recombination, or deletion), defined by Drake and Holland [71,72]. Specifically, the rate of mutation reflects the chance of a biochemical event caused by the replication machinery and its processing of the template, as a function of the intracellular environment. The mutation rate is typically reported for point mutations as a probability of occurrence per nucleotide copied; for deletions it is a probability per genome replication. For example, rates of deletion of 10^−8^-to-10^−6^ per replication have been estimated for the polymerase complex of the T7 phage [73]; larger deletions were lost at lower rates, based on measures of function recovery linked to deletion of different-length sequences engineered into the phage ligase.

In contrast to the mutation rate, the mutation frequency depends on both the mutation rate as well as the ability of the resulting DVG to enrich relative to other genomes in the population [72]. In short, the frequency of a given DVG is a population-level measure that is subject to Darwinian selection. The selection and persistence of DVGs in the lab and in nature depends in part on their ability to replicate and spread among cells as DIPs. An estimate of mutation frequency can be obtained by quantifying DIP levels relative to intact infectious particles descended from a single infected cell; in practice, this was done for populations of VSV isolated from small plaques [74], and estimated to be 10^−8^-to-10^−7^, a value that will reflect a combination of rates RNA recombination to form DVGs, and the fraction of DVGs that can be packaged and co-infect cells with intact virus to replicate and spread as DIPs. Similarly, a deletion frequency of about 10^−8^ was estimated for DVGs of phage T7 that deleted their gene encoding T7 RNA polymerase [75]. Such DVGs and their DIPs can reproduce faster than full-length genomes in recombinant host cells that supply this essential phage enzyme in trans [75]. This estimate of deletion frequency is based on quantification of DIPs as early as they can be detected in small-plaque populations of the phage. If the small plaques are permitted to expand, different DVG and DIP lineages emerge, each descended from the same ancestral phage genome; they enrich to different extents along different spatial directions, as the plaque expands radially [75]. By employing an engineered and artificial host cellular environment, this work revealed the emergence and evolution of a subset DVGs that would be otherwise undetected in natural infections.

Other aspects of DVG and DIP evolution have been revealed by other serial-passage or continuous cell and virus cultures. Such cultures have been shown to promote the emergence, enrichment, and displacement of longer DVGs by shorter higher fitness DVG variants. Such evolution of DVGs has been demonstrated by tomato bushy stunt virus (TBSV) during serial propagation on plants [76], by phage T7 during continuous culture on bacteria [77], and by VSV during serial-passage culture on mammal cells [14]. For TBSV, mechanisms of enhanced DVG stability and encapsidation efficiency did not account the fitness advantage of shorter DVGs [78]. Such results suggest the shorter DVGs gain a selection advantage over the longer DVGS, owing to their higher rates of replication. A library-based approach that generated and evaluated a broad range of DVGs lengths for different segments of influenza A supported this trend with more efficient replication by shorter genome segments [79]. However, in other cases of influenza A, selection processes can favor packaging of DVGs over intact genomes [80]. Efficiently packaged DI genomes have been found in mouse hepatitis virus [81], pseudorabies virus [82], and avian reovirus [83], which vary in competitiveness for encapsidation. Further phenomena include the emergence of super promoters that create an imbalance in the genome segments required for productive infection [84], or some combination of factors that do not require deletions to effectively replicate and interfere with normal virus growth [85].

The spatial distribution in cells of full-length viral genomes and their DVGs has been elucidated for Sendai virus by fluorescent in situ hybridization (FISH). In cells enriched with the full-length viral genomes, the viral genomes interacted with the cellular trafficking machinery and clustered in the perinuclear region. In contrast, in cells enriched in DVGs, the defective genomes showed no evidence of interaction with the trafficking machinery, and they were diffusely distributed around throughout the cytoplasm [86]. Resultantly, cells with full-length genomes produced both DVGs and fully infectious viral particles, while DVG-high cells poorly produced viral particles. This differentiation might be explained by the difference in intracellular localization of the DVGs.

More complex evolutionary dynamics of viruses and their defective genomes can exhibit co-evolution. The emergence of DVGs and DIPs can create an environment that selects for intact virus that resist interference. For example, mutants of intact VSV can resist DVG and DIP interference by mutations that impact genome replication and encapsidation [87]. Moreover, such DIP-resistant intact viruses further create an environment that promotes the emergence and enrichment of new variant DIPs; across hundreds of undiluted serial passage cultures, intact virus and their associated DIPs exhibit multiple rounds of co-evolution [88]. Other DIP-resistant intact viruses, including rabies [89], lymphocytic choriomeningitis [90], Sindbis [91], and West Nile virus [92], have the potential to exhibit similar co-evolutionary dynamics.

Finally, DVGs can evolve to cooperate in a manner that frees them from any dependence on intact virus for their growth. Serial-passage cultures of foot-and-mouth-disease virus (FMDV) at high multiplicities of infection enable the emergence and enrichment of DVGs and DIPs that are infectious by complementation; by co-infecting the same cell, they provide in trans the replication, encapsidation, and other functions needed to propagate their DVGs and their associated defective cooperating particles [93]. Such cooperation between separately packaged virus genome segments occurs in nature for multipartite viruses; the ecology and evolution of multipartite viruses has recently been reviewed [94]. The reverse of the functional segregation process, whereby separate virus-derived defective RNA segments recombine to form an intact fully infectious monopartite virus has also been demonstrated; DVGs of TBSV and a related cucumber necrosis tombusvirus that co-infected plant-derived protoplasts formed chimeric recombinants that were infectious in whole plants [95]. Similarly, engineered DVGs of Sindbis virus that were unable to replicate on their own in host cells could recombine with each other to form fully infectious virus [96]. The resulting cooperation between genetic elements that retain or lack replication and packaging functions have broader and deeper implications beyond virology, including key transitions in the origins of life [97].

#### 3.1.2. Measures of DIP Interference

The simplest models of DIP interference with virus growth assume single-hit behavior, where a cell co-infected with intact virus and at least one DI particle can only produce DI particle progeny [98,99,100,101], but experimentally observed interference behaviors are not quite so simple. The extent of interference depends on dose, the relative levels for standard, and DI particles that co-infect a cell. For example, higher concentrations of standard influenza A virus can partially rescue standard virus production during co-infection with DI particles [102]. Moreover, by interfering with the production of resources that are essential for standard virus replication, DI particles and genomes can, at elevated doses, inhibit their own replication.

Quantitative studies of VSV have provided evidence for complex feedback of interfering processes on DI genome and particle production. Using radioisotopes to metabolically label both DI and standard particles, density-based separation by ultracentrifugation, and quantification of both DI and standard particle populations, increasing inputs of DI particles were found to dramatically reduce production of standard particles, but the highest yields DI particles were under input doses that exhibited minimal interference with standard particle production [103]. Further quantitative studies using plaque-reduction assays and two-tiered titrations revealed highly non-linear virus and DI particle production with DI input dose [104], as shown in Figure 4. Specifically, for a fixed dose of virus and increasing DIP inputs, levels of standard virus production drop, as anticipated; greater numbers of DVGs can compete for replication and packaging resources, leading to fewer resources for standard virus production. However, at the highest levels of DIP inputs, standard virus production exhibits a partial recovery and increasing DIP dose causes DI particle yields to drop [104]. This phenomenon has been previously described as “interference of interference” [105]. A similar phenomenon has been described for engineered conditionally replicative or defective interfering HIV that “shoots itself in the molecular foot”; the DIP inhibits the virus so effectively that its own production is inhibited [106].

#### 3.1.3. DIPs *In Vitro*

DIPs have been found for many DNA and RNA viruses. DIP-forming DNA viruses include herpes simplex virus [107], pseudorabies virus [82], and Ageratum yellow vein virus [108]. RNA viruses that produce DIPs *in vitro* include vesicular stomatitis virus [109], Newcastle disease [110], measles [111], influenza [112], mouse hepatitis virus [113], and more recently, SARS-CoV-2 [114]. Defective viral genomes of SARS-CoV-2 form readily in culture [115], but it remains to be shown to what extent they interfere or interact with intact coronavirus growth.

#### 3.1.4. DIPs *In Vivo*

The discovery of DIPs for most viruses was initially considered to be irrelevant to natural infections owing to their origins *in vitro* [116], but in the last 15 years, preliminary evidence has emerged for DIPs *in vivo*. For example, short fragments of Dengue viral RNA containing only key regulatory elements for packaging at 3′ and 5′ ends and large internal deletions were found in serum from infected patients. Patient sera were used to infect mosquito cultures which developed identical RNA fragments, opening the possibility that the RNA fragments were transmitted as fully packaged DIPs. Transcribed *in vitro*, RNA corresponding to the *in vivo* samples were shown to be packaged into virus-like particles and transmitted over three passages in the presence of the wild-type virus. In preparations of Dengue virus with these short RNA fragments, yields of wild-type virus were reduced [117,118], providing support for interference and the most compelling evidence for DIPs *in vivo*. More recently, the same RNA from the serum was used to infect cell lines that support dengue virus growth, and again, infectious virus production was reduced; moreover, the RNA was shown to induce activity of RIG-I, MDA5, and interferon, consistent with the immunostimulatory activity characteristic of DIPs (see Section 3.1.5) [119].

RNA from nasopharyngeal samples of influenza A patients carried large internal deletions, overlap sequences at their 3′ and 5′ ends, and retained viral packaging signals [120] similar to RNA from DIPs generated *in vitro* [121,122]. Identical RNA sequences were found in two patients in the same contact group, pointing to the possibility of defective virus transmission. A more compelling case for natural DIPs of influenza A virus will follow from further studies; for example, interference with standard virus replication or growth has yet to be demonstrated, and DI particles rather than DVGs remain to be isolated and characterized. An inverse relationship was discovered between the number of defective viral genomes and the severity of an influenza A virus infection; specifically, clinical isolates showed a lower level of DVGs associated with a fatal infection and higher levels of DVGs in a mild case. These DVGs worked by inducing in their hosts a protective innate immune response [123]. It remains to be seen whether such DVGs are packaged and transmitted with viable virus.

Immunostimulatory defective viral genomes (iDVGs) of human respiratory syncytial virus (RSV) were found in human cells which triggered an antiviral immune response [124], characteristic of DIPs (see Section 3.1.5). However, the study focused on DVGs rather than DI particles; DIPs of *in vivo* RSV remain to be isolated and characterized.

Direct sequencing of a nasopharyngeal sample from a patient infected with Middle East respiratory syndrome (MERS) coronavirus revealed two virus variants with internal deletions that resulted in the truncation of viral proteins, and bioinformatic analysis suggested the variants were defective in packaging [125,126]. Functional tests for interference behavior have yet to be performed to provide evidence that these variants are DIPs. In the case of SARS-CoV-2, genomes that harbor large identical deletions were found in multiple patients who had only mild symptoms or were asymptomatic [127,128]. It remains to be seen what role, if any, DVGs may play in the severity of COVID-19.

#### 3.1.5. DIPs and the Immune Response

Recent reviews have summarized the role of DIPs in activating the immune response [69,116]. Specifically, DIPs induce type I interferons (IFN), which play a critical role in innate immunity. Mechanistically, the pattern-recognition receptor RIG-I preferentially binds with shorter genomes, such as the DVGs from DIPs, which then induces type I IFNs and other pro-inflammatory cytokines. In addition to activating immune responses, RIG-I activation can induce apoptosis; thus, DIPs have potential as anti-viral as well as anti-tumor therapeutics [129].

Beyond activating IFN associated innate immune responses, DIPs can also suppress IFN [130]. Eight strains of influenza A virus (IAV) were assayed in cell lines that hyper produce IFN, and their effects on IFN levels were quantified using interferon dose-response curves [131,132]. DIPs of strains which induced IFN were named IFN-inducing particles (IFPs) and particles which suppressed IFN were named IFN induction-suppressing particles (ISPs). The IFP activity appeared to be caused by the presence of a double-stranded RNA molecule, as particles with single-stranded RNA did not induce IFN; such effects of dsRNA on IFN induction had been well established [133,134]. IFN induction was enhanced by UV irradiation, which helped convert ISPs into IFPs and confirmed that the IFN activity is not dependent on virus replication or infectivity. IAV strains with deletions in NS1 were found to induce IFN at 20-fold higher levels than the parent stains; the inhibitory effects of NS1 on IFN induction are now well established [135,136]. These findings highlight how DIPs can activate or inhibit host defensive responses by their dsRNA structures or expressed anti-IFN functions.

Most recently, DVGs detected in nasal secretions of RSV have been associated with clinical patient responses to the virus. In children, DVGs were associated with a higher viral load and more robust pro-inflammatory response. In adults, however, the clinical response was based on the time at which DVGs were detected, rather than solely their presence. DVGs detected early in the course of infection were associated with mild disease, and DVGs detected later were linked to severe disease. Patients with DVGs had heightened expression of cytokines, including IFN alpha, which aligned with past studies linking induction of IFN to DVGs and DIPs [137].

Aside from their roles in triggering innate immune responses, work on Sendai virus has provided evidence of roles DIPs and DVGs play in adaptive immunity. Sendai stocks rich with DIPs were shown to induce dendritic cell maturation in human and mouse cells by helping upregulate cytokine activity. This pathway worked independently of IFN, and suggests possible applications of DIPs as vaccine adjuvants, stimulating dendritic cell maturation [138]. Subsequent studies showed that DIPs could upregulate the activity of pattern recognition receptors on dendritic cells which then stimulated T cell activation [139,140,141]. Also, Sendai DIPs were used as adjuvants for inactivated influenza A vaccines, where the DIP RNA enhanced the production of anti-influenza hemagglutinin specific IgG, showing that these DIPs exhibit broad adjuvant activity [140].

#### 3.1.6. DIPs as Antiviral Therapies

Owing to their ability to interfere with standard virus infections, DIPs have been proposed as potentially transmissible antiviral therapies. For example, a cloned DI virus of IAV with a large internal deletion, called 244 DI virus, has been administered intranasally to mice and found to protect against infections by several strains of IAV. In ferrets, intranasal administration of 244 DI reduced fever, weight loss, respiratory symptoms, and the infectious load of the standard virus relative to infected controls, providing further evidence that 244 DI virus can act as an effective antiviral [142]. A different DI virus derived from IAV, OP7, has exhibited strong interference when co-infected with IAV, based on a decrease in the infectivity of the released virions, supporting OP7 as a potential antiviral therapy [85]. Furthermore, therapeutic IAV particles have been engineered to spread DI genomic segments to divert normal IAV infection toward the production of non-infectious particles, with demonstrated protection against lethal virus in an animal model [143].

Engineered DIPs from human immunodeficiency virus (HIV) have been shown to reduce wild-type HIV replication [144,145,146,147]. Multi-scale models based on HIV data from sub-Saharan Africa suggested how so-called therapeutic interfering particles (TIPs) could lower HIV/AIDS prevalence by 30-fold in the next 50 years [148]. The proposed TIPs would be lentiviral vectors, which lack genes required to self-replicate but retain HIV packaging signals, so the vectors could move from hosts infected with HIV and outcompete the wild-type virus for resources. The difference between a normal DIP and these TIPs is that the TIPs would be engineered to have a basic reproductive ratio (*R_0_*) that is greater than 1; the TIP would generate more genomic RNA (gRNA) than the wild type virus, exploit viral resources made by the wild type virus, driving down wild type virus replication, disease progression, and transmission on large scale [149].

Recently, DIPs have been explored as therapeutics against flaviviruses. A production cell line used a combination of lentiviral and retroviral vectors to stably produce virus-free DIPs of dengue virus; following purification and concentration, the DIPs displayed antiviral activity in cells co-infected with dengue virus. Such production systems and DIP activities might well be realized for diverse viruses, providing a potential platform technology for DIP-based antiviral therapeutics [150]. For Zika virus, DVGs were computationally analyzed to pinpoint which DVG sequences would be most effective as TIPs. Sequences of DVGs that increased in frequency during consecutive serial passaging were identified and tested for interference activity against the wild-type virus. Then, the DVGs were engineered into VLPs, which displayed comparable interference, confirming their possible use as TIPs. The VLPs were used to infect mice and mosquitoes, reducing transmission of the wild-type virus by up to 90% in mosquitoes and reduced viral loads in the brains and ovaries of mice. Similar DVGs were found from passage cultures of West Nile and yellow fever virus, suggesting this methodology can be generalized to arboviruses, and potentially to others [151].

The COVID-19 pandemic has spurred the development of synthetic DVGs and DIPs against human coronaviruses, specifically SARS-CoV-2. Based on known DIPs of coronaviruses [115,152], a design incorporated 5′ and 3′ ends, and putative packaging signals from the SARS-CoV-2 genome [153]. The fragments were assembled in a frame, synthesized as DNA, inserted into plasmids, transcribed to form gRNA, and electroporated into cells infected with SARS-CoV 2. The DI genome was found to replicate about 3-fold faster than the wild type, while reducing the amount of wild type virus by about half in 24 h; a next step will be to evaluate the strategy in animal models. A more in-depth study designed therapeutic interfering particles based on mechanistic modeling, synthesis of TIPs that included 5′ and 3′ regions of the SARS-CoV-2 genome, packaging signals, and a fluorescent reporter. The TIPs were tested in cell culture, human lung organoids, and in hamsters. The engineered TIPs were found to inhibit SARS-CoV-2 replication by 10 to 100-fold in cells, and suppressed virus load by 10-fold in the lungs of hamsters, and reduced inflammation and severe disease when administered pre- or post-infection [154]. These encouraging results provide hope for eventual testing and optimization in human clinical trials.

Finally, engineered DIPs may contribute toward a new paradigm to promote public health: specifically, vaccines that reduce infectious disease owing to their broad protective effects [155]. For example, an enteroviral therapeutic interfering particle (eTIP1) based on polio DVGs triggered an antiviral state in the respiratory tract of mice that inhibited virus replication and protected against infection by enteroviruses, influenza, and SARS-CoV-2 [156]. The protection was achieved by administering eTIP1 within 24-to-48 h pre- or post-exposure to virus, and it was mediated by type I IFN signaling and a virus-specific neutralizing antibody response that persisted several weeks. Such strategies have the potential to protect against the emergence of virus variants owing to their broad antiviral effects.

### 3.2. Semi-Infectious Particles

Influenza A virus (IAV) typically exhibits particle-to-PFU ratios of 10-to-100 [157], so more than 90 percent of the particles in an IAV population are non-infectious. In a key experiment, IAV particles could initiate infections in single susceptible cells by starting gene expression, but most then failed to express one or more essential proteins; the cells also made no virus progeny, based on the lack of infection spread to nearby cells [158]. A small minority of cells infected by single particles exhibited productive infections. However, when cells were infected by multiple particles (at MOI 5), most cells were productive. Since cells infected by single or multiple particles exhibited either poor or efficient production, the invading entities were named semi-infectious particles (SIPs); see Figure 5.

Generally, SIPs differ from DIPs. SIPs do not interfere with standard virus production during co-infections, and they lack the large internal deletions that are genomic signatures for DIPs [159]. Furthermore, most SIPs of IAV appear to be fully intact; they carry each of the eight viral RNA genomic segments [160]. So why does only a small fraction of IAV particles productively infect cells?

High particle-to-PFU ratios can arise from the particles that lack essential functions for infection, such as DIPs, but also from SIPs, which one might expect to be fully functional based on their genome sequences, biomolecular composition, and structure. However, single-cell measures and quantitative models indicate an important role for stochastic or noisy processes in the mixed on/off behaviors of IAV SIPs and other viruses.

Common measures of average-cell behavior measured from a population of cells mask significant heterogeneity that can best be appreciated by measurements at the single-cell level [161,162,163,164]. In perhaps the most extreme case of masked behaviors, significant subpopulations of infected cells fail to make infectious virus progeny; specifically, infected cells that exhibit early viral gene expression fail to produce detectable viral progeny for 30 percent of VSV-infected cells [163], 80 percent of vaccinia-infected cells [165], and 90 percent for IAV-infected cells [158]. Furthermore, the remaining cells that produce infectious virus progeny exhibit yield distributions that vary from 10 to 1000-fold; such broad distributions have been observed for cells infected by phage [166], VSV [163], polio [167], and IAV [168].

Noteworthy effects of noisy behavior have been revealed by computational models of different intracellular processes associated with virus-cell interactions: stochastic gene expression in the lysis-lysogeny decision [169], heterogeneity in the internal levels of virus intermediates [168,170], and the sensitivity of infection to degradation of the entering virus genome [171]. Evidence for stochastic degradation of entering virus genomes for IAV combined experimental measures from single cells of viral RNA [168], viral proteins of IAV cells [158], and computational mechanistic modeling [168]. Further experiments and stochastic modeling of cytosolic diffusional transport have identified the time point of virus-endosome fusion and the associated diffusion distance for the release of the viral genome to the nucleus as a critical bottleneck for efficient virus infection [172]. To overcome the degradation of genomic segments, multiple virions are required for productive infection; specifically, for IAV, approximately 2-to-5 virions must enter a cell to render it productively infected [173]. In addition to overcoming a loss of genomic segments by degradation, studies of IAV show how dimerization of the viral RNA-dependent RNA polymerase may be needed to overcome host-specific barriers to viral RNA replication [174], providing evidence for a role of collective molecular interactions in essential viral processing.

The need for cooperation between virus particles is not limited to IAV. Elegant single-cell studies of vaccinia virus have combined nanoscale fluidic manipulation, detection of recombinant fluorescent virus particles, and atomic force microscopy (AFM) to dissect early stages of infection [165]. Based on the efficiency of single-particle inputs, one may predict efficiencies of two-, three-, and more-input particle behaviors; deviations from predictions indicate that the overall infection is cooperative. However, only 48 percent of surface-bound viruses enter, independent of the total number of particles bound, indicating entry is not cooperative. Furthermore, of those that enter, 80 percent are unable to directly detect gene expression, and less than 2 percent of surface-bound virus particles were able to complete the entire virus lifecycle and direct assembly of progeny virions [165].

Beyond the noisy or stochastic behavior, which is inescapable for single infected cells, other mechanisms may also contribute to the semi-infectious phenotype. For example, electron microscopy has shown that packaging of genomic segments for IAV can be incomplete [175], often omitting the segment encoding PB2 [176]. Furthermore, IAV has a high mutation rate [177], and mutations in multiple IAV genes can make it susceptible to shut-down by the innate immune system [178]. IAV mutations may also adversely affect how IAV gene products interact with its many essential host factors [179].

### 3.3. Non-Infectious Cell Killing Particles

Infectious virus particles typically kill their host cell as a byproduct of producing virus progeny. Particles that kill their host cell but fail to make detectable progeny have been called non-infectious cell killing particles (NiCKPs), as shown in Figure 6. In general, such particles may be quantified by the clonogenic assay (Section 2.3), where cells that are not killed produce colonies that can be readily quantified.

#### 3.3.1. NiCKP Characterization

Evidence for NiCKPs was based on differences between measures of cell-killing and plaque-forming by particles of VSV. The CKPs that failed to produce detectable progeny by plaque assay were “defective cell killing particles” or NiCKPs, which were produced at 5 to 9-fold higher concentrations than PFPs [61]. Furthermore, NiCKPs and DIPs both failed to make virus progeny, but DIPs also failed to kill cells, and DIPs failed to interfere with host cell killing by NiCKPs. Similar behaviors have been demonstrated for NiCKPs and DIPs of IAV [180]. A single NiCKP is sufficient to kill a cell [55], and UV inactivation studies indicate the theoretical target of UV inactivation differs in size for different IAV particles. More specifically, normal infectious particles have a UV target of about 13,600 nucleotides (nt), NiCKPs have a UV target of about 2400 nt, consistent with one of the polymerase subunit genes, and DIPs have a UV target of about 350 nt, consistent with the smallest defective viral genomes associated with interference [180]. Finally, assays for infectivity, interference, and cell killing were combined to characterize the dynamics of IAV populations during high-multiplicity passages; an initial population of pure infectious particles dropped more than 100-fold during the first passage as it was replaced by DIPs (68.5 percent) and non-infectious CKPs (31 percent). During second and third passages, DIPs continued to enrich (above 90 percent) at the expense of non-infectious CKPs (below 10 percent) and infectious particles (~0.01 percent), while total particle counts remained relatively stable and high, above 10^9^, across passages [180].

#### 3.3.2. Particle Fitness and Virulence

Although descriptions of virus particles as infectious or cell-killing suggest all-or-nothing behaviors, such descriptions are simplifications for a continuum of behaviors. The fitness of a virus under specified conditions typically refers to its replicative ability, while its virulence refers to its capacity to kill cells [181]. The fitness and virulence of a virus depend on its culture conditions. For example, when a clone of foot-and-mouth disease virus (FMDV) was passage cultured at large population numbers, high-fitness high-virulence viruses resulted; however, subsequent plaque-to-plaque passages produced significant losses in fitness with little loss in virulence [181]. Such low-fitness high-virulence FMDV particles are analogous to the non-infectious CKPs described for IAV [180]. Likewise, studies of point mutations on the effects of FMDV fitness and virulence give an indication of how these phenotypes are encoded; fitness can be affected by mutations in any region of the genome, while virulence appears to be localized to a subset of viral genes. As a result, virulence can be more robust than fitness to the effects of deleterious mutations [181]. Studies of fitness and virulence in other viruses exhibit a variety of behaviors; viral fitness and virulence are positively correlated but with noteworthy exceptions for VSV [182], and they appear to lack correlation for infections in plants by *Tobacco etch potyvirus* [183].

#### 3.3.3. Application of NiCKPs

Virus-like particles that span a broad range of virulence or cell-killing may have useful applications in human health. Non-infectious CKPs, which have low fitness but remain virulent, may be useful as therapeutics where cell killing is desirable but not infection-spread, as in oncolytic therapies to treat cancer. For example, engineered highly attenuated VSV, which makes few progeny and small plaques, selectively infected and killed human gliomas implanted in SCID mice [184]; oncolytic therapy by VSV has been tested in the United States in phase I clinical trials [185]. Others have argued for oncolytic strategies that spread rapidly in the host, owing to their high fitness, but are minimally virulent [182]. Other factors beyond the level of virus fitness and virulence, including the extent of innate immune activation in healthy and targeted host cells and tissues, will also be important in the design of oncolytic therapies.

## 4. Discussion

**Particle-to-PFU ratios.** What significance should one assign ratios of particle-to-PFU or genome-to-PFU that are far greater than one? The ability to quantify such ratios require total particle or genome levels, in addition to PFUs measured from the same sample. Such multiple measures are most often performed on samples from “clean” laboratory cultures of cells and virus rather than from “dirty” or uncultivated environmental samples. Such cultures also typically use transformed cell lines to host the infection, as they can give robust virus titers or enable easy visualization and quantification of infectious particles by plaque counting. The virus strains are often far removed from their natural counterparts, having been selected over multiple generations of passage cultures in the lab to yield robust titers or easily visualized plaques. The broad ranges of reported particle-to-PFU ratios also reflect a lack of reference ratios and standardization [186]. Nevertheless, we believe that the utility of such ratios is not so much in their specific numerical value, but instead, when such ratios are larger than one, they underscore two features that may well hold for viruses and their hosts in nature: (i) non-infectious virus-like particles easily arise as byproducts of infection, and (ii) such particles can exhibit diverse biological activities.

**Defective viral genomes in nature.** We have highlighted compelling examples of diverse biological activities of virus-like particles and DVGs *in vivo* or from natural and clinical isolates. These include semi-infectious particles, which must infect the same cell with multiple particles in order to give a productive infection for influenza A virus in mice [158]. Furthermore, the viral genomes with similar or identical defects in essential genes have been isolated from human patients and mosquito vectors at different geographical locations for Dengue virus infections, providing evidence for their long-term transmission over space and time [117]. More recently, the presence of DVGs arising in humans has been associated with induction of innate immune responses [124] and linked to different extents of disease severity for young and old patients of respiratory syncytial virus [137]. Finally, it is noteworthy that DVGs of polyomavirus, a double-stranded DNA virus, have been associated with higher viral loads in clinical samples [187]. We anticipate that deep sequencing and single-molecule sequencing of patient and environmental isolates will, in the coming years, reveal still further examples of DVGs associated with diverse clinical outcomes. An ongoing challenge will be to characterize the potentially multiple biological activities of such DVGs, as well as their underlying mechanisms. Such mechanisms may well act across scales from molecules to particles, and further to multiple-particle populations.

**Engineered study of viruses and virus-like particles.** Engineered experimental systems can enable the study of virus-cell behaviors at a deeper level than would be practical or ethical for natural or patient infections. For example, engineering includes design, synthesis and application of reporter genomes. Single-reporter viruses can be used to infect cells at low MOI, combined with fluorescent-activated cell sorting (FACS) to isolate single cells infected by single virus particles, which can be further studied to study infections start viral gene expression but fail to produce virus, as well as distributions of yields from productive infections [163]. Engineered dual-color reporting from virus and DIPs can enable quantification of virus-encoded and DIP-encode gene expression across populations of single cells under different conditions of co-infection [188]; micro-well technologies and associated image acquisition and analysis pipelines can facilitate tracking of such single-cell behaviors over time [162]. Alternatively, engineering may involve the coupling of wet-lab infection “titrations” with mathematical or computational modeling; effects of DIP dose on the yields virus and DIP activity from co-infected cells have revealed non-intuitive behaviors, especially at high DIP doses, and mathematical modeling of the data have suggested single DIPs of VSV cannot fully inhibit viable virus production [104]. Finally, engineering of artificial environments to probe isolated single cells over time, when compared with more common populations of cultured cells, can reveal subtleties of DIPs [162]; co-infection of cells in populations are less inhibited by DIPs than their single-cell counterparts, suggesting nutrient or signaling effects within cell populations that temper the inhibitory effects of DIPs on viral gene expression and growth [189].

**How can a virus be “semi-infectious?”** Mathematical modeling of single cell behaviors has highlighted how noisy or stochastic degradation of virus genomes can cause otherwise identical infected cells to either produce virus or not [171]. More mechanistic and complex computational models have shown how stochastic synthesis of transcripts, proteins, and genomes during single-cell infections may contribute to the broad distribution of virus yields [170], which have been observed experimentally [163,166,190]. Such models provide a plausible contributing factor to the phenotype of semi-infectious particles [159]. More broadly, the challenge of particle-associated infection has been well articulated by Klasse; “all-or-nothing assumptions about virion infectivity are flawed and should be replaced by descriptions that allow for spectra of infectious propensities” [191]. In other terms, the outcomes of a virus-cell interaction can exhibit a broad distribution of infection-associated behaviors, including failure to make virus progeny.

**Returning to nature.** Transformed cell lines commonly used to culture viruses are often defective in innate immune signaling and suppression of virus growth; these include, for example, Vero [192,193], BHK [194,195], HEK293T [196], and HeLa [197]. Al-though cases are known where differences between viral genome replication, synthesis and processing of viral proteins, and viral shutoff of host cell processes on such transformed cell lines are comparable to their behavior in primary cells [198], such studies are rare; comparisons of virus growth and infection spread in culture highlight greater inhibition, where host cells retain innate immune signaling. To what extent virus propagation on primary cells and tissues gives rise to the diversity of virus-like particles generated from culture on transformed cells remains an open question.

To culture primary cells is technically challenging, and their infections by viruses tend to be less productive and highly variable. However, it is plausible that the low productivities and variability reflect features of infection that are a step closer to their behaviors in nature. Other more natural and challenging to implement culture environments include conditions that promote primary cell differentiation. For example, air–liquid interface (ALI) cultures of primary human airway epithelial cells promote their differentiation to create a pseudostratified epithelium; goblet and mucus-producing cells are present, as well as functional cilia, and the epithelium is susceptible to infection by human rhinovirus-C [199]. The fields of regenerative medicine and drug testing, which tailor differentiation of human induced pluripotent stem cells (iPSCs) toward tissue-like cells, offer potentially more natural and controlled environments to study virus growth and infection spread behaviors. For example, iPSCs have been differentiated to create hepatocyte-like cells that support the full life cycle of hepatitis C virus, including inflammatory host responses to infection [200]. Similar approaches have been used to create human neural progenitor cells for study of the Zika virus [201] and human 3D lung bud organoids for the study of respiratory syncytial virus [202]. As these technologies mature and become more widely used, they may help reveal facets of virus and virus-like particle interactions during growth and spread that are absent from cultures on transformed cell lines.

**Conclusion.** Large particle-to-PFU ratios measured for many viruses suggest that the vast majority of virus-like particles are unable to productively infect their host cells. However, despite being “dead”, virus-like particles can be very much alive in other facets: parasitizing the resources within host cells that intact viruses establish for growth, interfering with normal virus growth, activating or inhibiting innate immune signaling of their host cells, killing their host cells, and teaming with other “dead” viruses to produce infectious progeny. Such understanding of virus-like particles has been largely gleaned from studies at the level of particles and host cells. A grand challenge remains to understand how such functions and activities impact the development, severity, transmission, and persistence of infectious disease in their plant, animal, and human hosts.

## Figures and Tables

**Figure 1 viruses-14-00383-f001:**
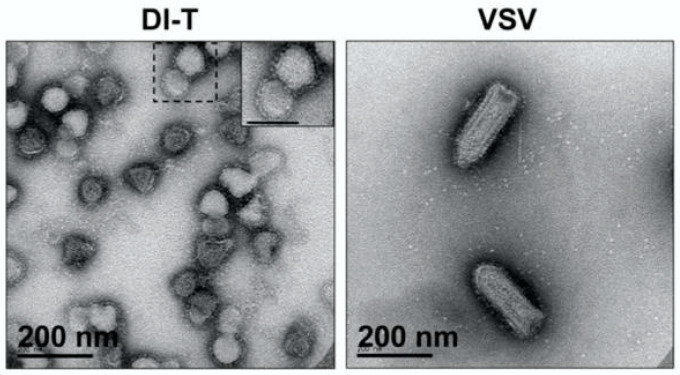
Transmission electron microscopy images of vesicular stomatitis virus defective interfering (DI-T) and standard (VSV) particles. DI-T particles are truncated relative to full-length VSV particles. Image adapted from [31].

**Figure 2 viruses-14-00383-f002:**
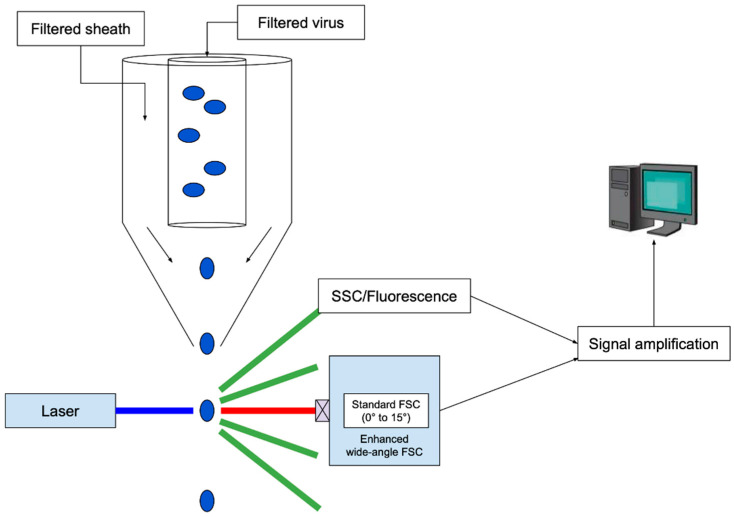
Counting of virus-like particles by flow virometry. The apparatus employs a flow-focusing stream to align particles as they move single file through the beam of a powerful laser, followed by enhanced wide-angle FSC detection. The standard FSC signal used in conventional cell counting is blocked, and the wide-angle FSC signal is enhanced by setting a higher threshold for detection, reducing noise, and thereby enabling sensitive detection of nano-scale particles. Image adapted from [45].

**Figure 3 viruses-14-00383-f003:**
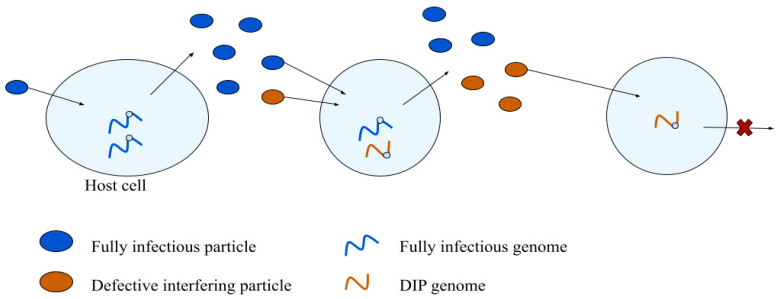
Defective interfering particles (DIPs): emergence and biology. DIPs arise from normal virus infections (**left**), and they amplify during cell co-infections with fully infectious particles (**middle**). DIPs alone are unable to productively infect cells (**right**).

**Figure 4 viruses-14-00383-f004:**
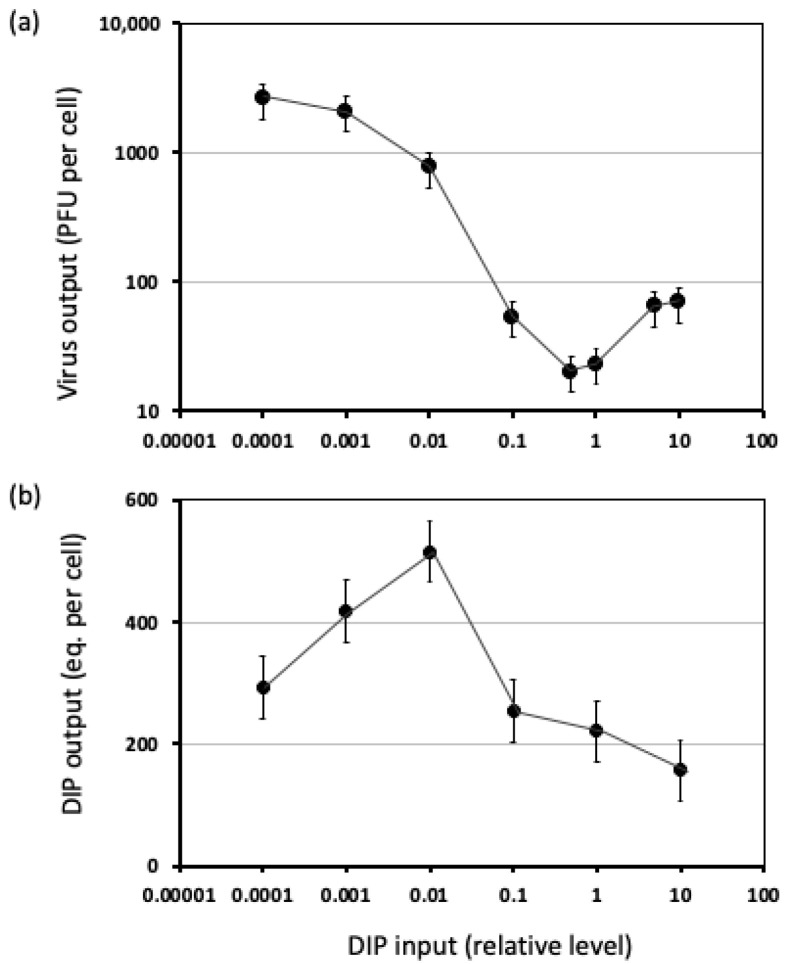
Production of (**a**) standard virus and (**b**) DI particles by co-infected cells is a complex function of DI particle inputs. All cells were co-infected with standard virus (MOI 20) and different input levels of DI particles. Levels of standard virus and DI particles were determined by plaque assay and plaque-reduction assay; adapted from [104].

**Figure 5 viruses-14-00383-f005:**
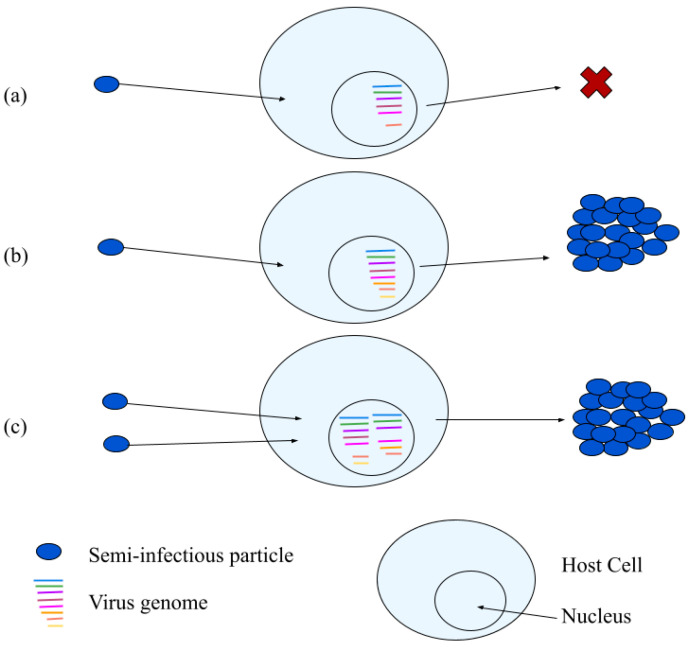
Three potential fates of semi-infectious particles. (**a**) A particle infects its host cell but at least one essential gene fails to be expressed, so the cell makes no progeny, (**b**) a particle infects its host cell, all essential genes are expressed; the cell make progeny, and (**c**) two or more particles co-infect a host cell, gene or functional deficiencies are overcome by complementation, and the cell makes virus progeny.

**Figure 6 viruses-14-00383-f006:**
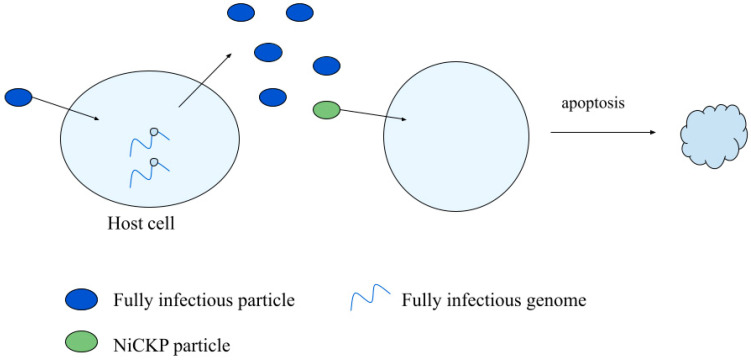
A non-infectious cell killing particle can trigger apoptosis in a cell.

**Table 1 viruses-14-00383-t001:** Virus particle quantification techniques and their characteristics.

Technique	Detection Type	Counting	Throughput	Detection Limit	References
Transmission electron microscopy	Physical virus particle	Manual and automated	100 particles per grid	≥10^7^ particles/mL	Borries (1938) Timm (2014) Blancett (2017) Roingeard (2019)
Epifluorescence microscopy	Fluorophore-labelled virus particle	Manual and automated	100 particles per grid	≥10^7^ particles/mL	Chen (2001) Ortmann and Suttle (2009) Parveen (2018)
Tunable resistive pulse sensing	Physical virus particle	Automated	10,000 particles per sec	10^7^–10^10^ particles/mL	Akpinar (2015) Yang (2016)
Flow virometry	Fluorophore-labelled virus particle	Automated	2000–6000 particles per sec	10^5^–10^9^ particles/mL	Rossi (2015) Zamora (2017)

**Table 2 viruses-14-00383-t002:** Virus particle quantification techniques based on biological functions and characteristics.

Technique	Detection Type	Counting	Time	Units	References
Plaque assay	Infectious virus particle	Manual	2–14 days	pfu/mL	Baer and Kehn-Hall (2014)
End-point Dilution assay	Infectious virus titer for 50% CPE	Manual	Varies depending on infection time of virus	TCID_50_/mL	Flint et al. (2004) Reed and Muench (1938)
Clonogenic assay	Cell-killing particle	Manual or Automated	Based on incubation time for visible colonies (1–3 weeks for eukaryotes)	CKPs/mL	Ngunjiri et al. (2008) Franken et al. (2006)

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
