# Peer review of "Virus-like Particles: Measures and Biological Functions"

_viruses, 2022, doi:10.3390/v14020383_

Round 1

Reviewer 1 Report

The authors provide in their review article with the title “Virus-like-Particles: Measures and biological functions” an excellent summary of the importance of virus-like particles (VLPs) from a virologist's point of view. They summarize the specific terms and descriptions of the rising forms of VLPs, define them and put them in the context on the current knowledge of their functions. On the other hand they open the view for the many open questions of VLPs functions and our understanding of their roles in virus infections and potential in antiviral therapies and vaccine development. The article is very well written and figure illustration very clear. The authors provided an outstanding collection of references on original articles and summarize the conclusions and point to future research directions on this topic.   

Specific minor: In the line 97 and 606 there are two dots at the end of the sentence. 

Author Response

Specific minor: In the line 97 and 606 there are two dots at the end of the sentence.

These typos have been corrected.

Author Response

1) It surprises me that fluorescence in situ hybridization (FISH) analysis is not mentioned in the review. Since defective viral genomes are often involved in defective interfering particles the visualization of such genomes by FISH is interesting. It could be mentioned under 3.1.4. DIPs in vivo.

We thank Reviewer 2 for highlighting the application of FISH in characterizing the spatial distribution of defective and full-length viral genomes in cells. We now cite in line 365 the relevant study by Genoyer and Lopez, 2019.

2) I suggest the authors incorporate the following very recent paper also in their review:

Library-based analysis reveals segment and length-dependent characteristics of defective influenza genomes (plos.org) This could be done at the end as a novel research approach that can be more broadly applied than only influenza, but possibly also at other positions.

As suggested by Reviewer 2, we have now cited this work on line 353.

3) Line 253: the abbreviation PFP first appears here and is not defined. I assume it is plaque-forming particle.

This abbreviation has now been defined.

4) Line 372: please insert the word ‘genome’ into “separately packaged virus components” to make it more clear and read “separately packaged virus genome segments”

This change has been made.

5) Line 461: space before full stop....

This change has been made.

6) Line 519: ‘wildtype’ is written without space, which is not consistent with other parts of the review.

This has been fixed, it now reads “wild type.”

7) Line 606: double full stop at end....

This typo has been corrected.

Reviewer 3 Report

Bhat et al. review on propagation-incompetent virus-like particles of different virus families that arise both in vitro and in vivo. They present extensive technical details on assays to quantify total virus particle counts and infectious virus particle titers. The major focus of the review are defective interfering particles that may be used as an antiviral agent. They further report on semi-infectious particles and non-infectious cell killing particles, their origin, implications on virus evolution and potential clinical use.

The review is well written and comprehensive. I only have one comment.

As the major focus of the article are defective interfering particles, the authors could have elaborated more on the mechanisms of DIP interference. For instance, in line 333 and 390-391, it is just shorty mentioned that shorter genomes replicate faster, depleting resources of the competitor virus. Mechanisms of interference at the genome packaging step are not mentioned at all. For instance, for influenza A virus, it was reported that defective interfering genomes can specifically interfere with their parental, full length genome at the assembly step. What about other virus families?

Author Response

As the major focus of the article are defective interfering particles, the authors could have elaborated more on the mechanisms of DIP interference. For instance, in line 333 and 390-391, it is just shorty mentioned that shorter genomes replicate faster, depleting resources of the competitor virus. Mechanisms of interference at the genome packaging step are not mentioned at all. For instance, for influenza A virus, it was reported that defective interfering genomes can specifically interfere with their parental, full length genome at the assembly step. What about other virus families?

As suggested by Reviewer 3, we have now expanded the description of mechanisms of interference that include packaging and assembly. These include packaging and assembling of defective interfering genomes of influenza A, mouse hepatitis virus, pseudorabies virus, and avian reovirus; associated new references are also cited (lines 354-356).